evolution/behaviour/immunology

lifespan, mortality, parasite infection, social interactions, metabolic rate

**Author for correspondence:**
Susanne Foitzik
e-mail: foitzik@uni-mainz.de

# Extreme lifespan extension in tapeworm-infected ant workers

Sara Beros[1], Anna Lenhart[2], Inon Scharf[3], Matteo Antoine Negroni[2], Florian Menzel[2] and Susanne Foitzik[2]

[1]Max Planck Institute for Biology of Ageing, Cologne, Germany
[2]Institute of Organismic and Molecular Evolution, Johannes Gutenberg University, Mainz, Germany
[3]School of Zoology, Faculty of Life Sciences, Tel Aviv University, Tel Aviv, Israel

 
IS, 0000-0002-8506-7161; MAN, 0000-0003-0177-5783;
FM, 0000-0002-9673-3668; SF, 0000-0001-8161-6306

Social insects are hosts of diverse parasites, but the influence of these parasites on phenotypic host traits is not yet well understood. Here, we tracked the survival of tapeworm-infected ant workers, their uninfected nest-mates and of ants from unparasitized colonies. Our multi-year study on the ant *Temnothorax nylanderi*, the intermediate host of the tapeworm *Anomotaenia brevis,* revealed a prolonged lifespan of infected workers compared with their uninfected peers. Intriguingly, their survival over 3 years did not differ from those of (uninfected) queens, whose lifespan can reach two decades. By contrast, uninfected workers from parasitized colonies suffered from increased mortality compared with uninfected workers from unparasitized colonies. Infected workers exhibited a metabolic rate and lipid content similar to young workers in this species, and they received more social care than uninfected workers and queens in their colonies. This increased attention could be mediated by their deviant chemical profile, which we determined to elicit more interest from uninfected nest-mates in a separate experiment. In conclusion, our study demonstrates an extreme lifespan extension in a social host following tapeworm infection, which appears to enable host workers to retain traits typical for young workers.

## 1. Introduction

Reproductive and task division of labour have been long acknowledged to be hallmarks of eusociality in insect societies. The queen specializes in reproduction and the workers collectively perform all other tasks, including caring for the queen and her offspring, building and defending the nest, and

foraging for food [1]. An intriguing feature of many social insects, and of ants in particular, is the stark divergence in lifespan between female castes, which are typically not genetically determined [2]. Many ant queens exhibit extraordinarily long lifespans of several decades. They remain fertile during their lifetime spending it almost exclusively inside the nest tended by their worker daughters [3]. By contrast, ant workers—being sterile—exhibit much shorter lifespans of only a few weeks, months or rarely years [4–6]. These sterile workers take over all chores in the nest, and as they grow older, they switch from inside work such as brood care to the much riskier outside tasks [7].

The social environment is known to improve the health and survival of animals typically living in groups [8–10]. In social insects, this is especially apparent for queens, whose long lifespans can be in part explained by the high levels of social care they receive from their workers and the reduced extrinsic mortality due to the safe environment in the nest [9]. Despite these obvious benefits of sociality, a social lifestyle with the close proximity of genetically similar group members also provides favourable conditions for parasites to spread and thrive [11,12]. Parasites typically reduce the fecundity and survival of their insect hosts due to their reliance on their hosts' resources [13]. Social insects serve as hosts for diverse parasites and workers leaving the nest to forage for food are frequently exposed to parasites [12–15]. Infections in social insect workers often lead to behavioural changes and can even accelerate the behavioural maturation of young workers [16–18]. This highlights the importance of parasites for the phenotypes of social insect hosts and reveals how social traits can be intertwined with parasite-induced alterations.

Parasite infection generally incurs fitness costs to their hosts, such as a slower development, reduced survival or a lower fecundity [19]. Surprisingly, some parasites extend the lifespan of their hosts, for instance by interfering with the fecundity–longevity trade-off by reducing the reproductive success of hosts up to complete sterilization [20–22]. We have previously demonstrated increased survival over a few weeks of tapeworm-infected workers of the small Central European ant *Temnothorax nylanderi* [23]. Ants of this species serve as an intermediate host to the trophically transmitted tapeworm, *Anomotaenia brevis* [23]. Infected workers do not show reduced reproductive potential compared with their uninfected worker sisters; on the contrary, they develop their ovaries even more strongly when the queen is removed [24]. Ant workers get infected during the larval stage when fed with tapeworm eggs [25], which develop within the ants to parasitic cysticercoid larvae. A single ant can be parasitized by as many as 70 cysticercoids (T. Sistermans 2020, unpublished data; [26]), that reside in the haemocoel of their abdomen. The complex life cycle of *A. brevis* is completed when woodpeckers prey on parasitized ant colonies that live in cavities of sticks or acorns on the forest floor. Then, inside the bird's gut, the cysticercoids develop into adult tapeworms [27]. Next to increased survival, tapeworm infection leads to a multitude of phenotypic changes in infected workers, which are easily identified by their yellow, less sclerotized cuticle compared with their brown nest-mates [28]. Infected workers are less active, stay on the brood pile and exhibit reduced flight behaviour [23,26]. These behavioural changes are also reflected in an altered gene expression in the brain and abdomen [29,30].

The main focus of our multi-year study was to investigate the long-term consequences of *A. brevis* infection on *T. nylanderi* workers. We were interested in how long infected workers can live or whether their reported increase in survival might be due to traits more typically expressed by younger ants. We therefore tracked worker and queen survival in parasitized and unparasitized ant colonies over 3 years until over 95% of the uninfected workers had died. The factors that directly or indirectly regulate lifespan in social insects are still poorly understood. In particular, the role of social behaviours of group members on longevity has hardly been studied so far. In our focal species, infected workers are known to be fed more often than their nest-mates [26]. We thus were interested to gain insights into whether ants in parasitized societies receive more social attention depending on their infection status or caste. We further examined whether the deviant cuticular hydrocarbon profile of infected workers is more attractive to their nest-mates, which could explain why they receive more attention [28,31]. Physiological traits are known to be associated with the pace of life and could therefore be good markers of longevity [32–34]. We thus finally analysed the metabolic rate and lipid content of workers and queens in parasitized and unparasitized colonies.

## 2. Methods

The ant colonies in this study were collected in forested areas close to Mainz-Wiesbaden, Germany, from three different sites: (i) 50°00′36.4″ N, 8°10′47.3″ E; (ii) 50°02′29.4″ N, 8°02′46.6″ E and (iii) 50°05′42.8″ N, 8°09′55.1″ E.

## 2.1. Survival of uninfected and infected workers and queens

We noted the survival of workers and queens of different age groups from parasitized and unparasitized colonies for 3 years (1110 days, start: 22 September 2014, end: 6 October 2017). We collected 8 and 22 parasitized *T. nylanderi* colonies and 9 and 19 unparasitized colonies in September 2013 and April 2014, respectively. All colonies were queenright and comprised between 22 and 245 workers ($121 \pm 58$ workers: mean $\pm$ 1 s.d.). Colony size (i.e. the number of workers) did not differ between parasitized and unparasitized colonies (Wilcoxon test: $W = 429.5$, $p = 0.89$). *Temnothorax* ants display a synchronized annual reproductive cycle. During four weeks in summer, all new workers and sexuals emerge from the pupae. We took advantage of this synchronized emergence in mid-September 2014 to differentiate between young and old workers. Young workers—so-called callows—are easily distinguishable from older ones by their light, not yet fully sclerotized cuticle. These workers focus on brood care and are henceforth referred to as nurses. Workers leaving the nest to forage for food—so-called foragers—are usually older and were at the time of collection likely to be 1 year or older [31,32]. *Temnothorax nylanderi* ants are brownish with a characteristic dark abdominal stripe that is visible in callows, but is missing in infected workers with their unpigmented, yellow cuticle [25,26,28].

Within a day of emergence, we wire-marked (ELEKTRISOLA, 0.025 mm) five nurses ($n = 285$) and concurrently five foragers ($n = 285$) in each colony. All nurses and foragers were identified to be uninfected based on their body coloration (e.g. brownish, with abdominal stripe) in this and all following experiments. In parasitized colonies, we additionally marked between one to five infected workers (depending on availability), which were identified by their unpigmented cuticle ($n = 98$). To be able to distinguish young from old workers, wire-markings were unique for nurses and foragers within each colony, but were randomized between colonies. Ant queens can be easily distinguished based on their distinct morphology (e.g. larger body and structured thorax) and were not wire-marked. As *T. nylanderi* is strictly monogynous [24], queens were likely to already be a few years old at the beginning of the observations, as they had successfully established a colony. At the onset of our observations, all infected workers and selected foragers were at least 1 year old, whereas callows were only 10 days old.

The ant colonies were maintained in observation nests consisting of two glass slides separated by a piece of plexiglas providing a cavity of $4.9 \times 1.1 \times 0.3$ cm. These observation nests were placed in three-chambered boxes with a moistened plaster floor and kept in climate chambers, set to temperatures and photoperiods typical to the season (December–February: 10°C : 5°C day : night (DN) temperature and 10 h : 14 h light : dark (LD) period; March–May: DN 20°C : 15°C and LD 12 h : 12 h; June–August: DN 25°C : 18°C and LD 12 h : 12 h; September–November: DN 18°C : 13°C and LD 12 h : 12 h). We recorded the survival of all marked workers and the queens at 10-day intervals. The day of death was recorded as the last day the ant was observed alive. During the 3 years, 16.8% (48 of 285) of marked young workers, 21.0% (60 of 285) of marked foragers and 15.3% (15 of 98) marked infected workers disappeared, with no corpses found. For these individuals, the day of disappearance was entered as the day of death. We found the corpses of all queens who died. On each observation day (i.e. every 10 days), colonies were fed with pieces of crickets and a droplet of honey, except during hibernation (December–February), when we provided colonies with a droplet of honey at every second observation. Water was offered ad libitum throughout the entire time. Ant survival was examined using Kaplan–Meier analyses allowing for right-censored data, that is the number of days until death per individual. The log-rank Mantel–Cox test was employed to determine statistical differences and groups were compared for their hazards. All statistical analyses were performed in the Graphpad Prism v. 8.3.0 software.

## 2.2. Worker and queen metabolic rate, body mass, lipid content and social care

We collected an additional set of 14 parasitized and 14 unparasitized colonies in April–May 2018. All colonies were headed by a single queen and contained $69 \pm 33$ workers (mean $\pm$ 1 s.d.). Parasitized and unparasitized colonies did not differ in colony size (i.e. number of workers or brood, Wilcoxon tests, both $p > 0.7$). Infected workers and queens were easily recognized based on colour or morphology (see above). As all uninfected workers were non-callow adults (i.e. sclerotized cuticle), nurses and foragers were identified by their behaviour and location in the nest: nursing the brood inside the nest or foraging outside [24,35]. We cannot provide the exact age information about these ants. However, as colonies contained more than 20 workers, queens were likely to be a few years old. All workers were more than 10 months old, as the experiment began in June before the emergence of the new annual worker generation, which emerges in late summer. In *Temnothorax*, foragers are generally older than brood care workers [36,37].

We first measured the $O_2$-consumption of workers and queens from each parasitized and unparasitized colony. Measurements were taken with the MicroRespiration system from UNISENSE (Denmark), following their protocol. Ants were isolated from their colony and individually placed in a micro-respiration chamber ($v = 0.448$ ml) and sealed with agar and paraffin oil. The glass chamber was transferred to a water bath at a constant temperature of 23°C. A thin capillary in the chamber lid served as an oxygen microsensor to measure the $O_2$-consumption. Real-time $O_2$-consumption was recorded for 10 min and viewed using the free software SensorTraceBasic v. 3.0.200. All ants were weighed directly after the measurement (accuracy of 1 µg; PESCALE Wägetechnik). We calculated the respiration rate using the linear section of the $O_2$-consumption slope (from minute 5 to minute 10), adjusted for the live body mass (mg). The variable we used was the slope of $O_2$-consumption ($\mu mol\, l^{-1}$) plotted against time (s), divided by the ant mass (mg) and multiplied by the chamber volume (ml), hereafter 'metabolic rate'.

Following the respiration measurements, the ants were randomly marked with coloured wires and returned to their colony. On the following day, we conducted 20 behavioural scans of each focal individual and noted down whether or not an individual received social care from their nest-mates (i.e. being groomed, fed, carried or antennated). Thereafter, we calculated the rate of social care occurrences from all 20 observations. We also noted down whether we observed active begging behaviour for food. The next day, all marked ants were individually frozen at −20°C. To extract their lipids, they were placed in a chloroform/methanol mixture (2:1, v/v) for 24 h [28]. Nonadecanoic acid (C19:0) was added as the internal standard (20 µl in DCM/MeOH, 2:1 v/v; 0.2 mg ml$^{-1}$). The extracts were then fractionated in Chromabond SiOH columns (1 ml; Macherey-Nagel). Each column was conditioned with chloroform and hexane, and the lipids were eluted with chloroform. The samples were dried under a nitrogen stream and dissolved in 250 µl of a 2:1 dichloromethane/methanol mixture (v/v). Lipid extracts were analysed with a 7890A gas chromatograph (Agilent) coupled to a 5975C mass-selective detector (Agilent). The oven programme started at 60°C for 1 min, then increased by 15 K min$^{-1}$ to 150°C, followed by an increase of 3 K min$^{-1}$ to 200°C and finally a ramp of 10 K min$^{-1}$ up to 320°C, where it remained constant for 10 min. Peak areas were integrated manually using the Agilent software MSD Chem Station E.02.02. The data were then exported to MS Excel and manually aligned. The fatty acids had chain lengths between $C_{12}$ and $C_{20}$ and were identified based on diagnostic ions, retention time and the molecular peak. We calculated the total lipid content from the quantity of the internal standard and the quantity of all fatty acids together, and divided it by live body mass (mg) to obtain the relative lipid content.

The datasets of metabolic rate, body mass (both square-root transformed) and lipid content were separately analysed using linear mixed models (LMM). The relative frequency of social interactions was assessed using general linear models (GLMM) following binomial distribution with a logit-link function. We examined whether ants differed in their metabolic rate, body mass, lipid content and the frequency of social care depending on the parasitism status of the colony, their own caste and infection status. Hence, each model (LMMs and GLMM) incorporated colony parasitism (parasitized/unparasitized) in interaction with the category of the individual ant (infected/nurse/forager/queen) as fixed predictors and colony identity as the random factor. We applied a backward stepwise selection procedure for model selection ($\alpha = 0.05$). Neither metabolic rate, body mass, lipid content nor the frequency of social care was impacted by the parasitism status of the colony (see results). We thus only focused on parasitized colonies, analysing whether and how the metabolic rate (LMM), body mass (LMM), lipid content (LMM) and the frequency of social care (GLMM) differed among infected workers, nurses, foragers and queens. The individual ant category served as a single explanatory variable and colony identity was included as the random factor. Pairwise comparisons between groups were obtained by re-running the models by relevelling factor levels. Models were run in R v. 3.3.2 using package lme4 (commands lmer, glmer) [38].

## 2.3. Response to cuticular hydrocarbons of infected and uninfected workers

Chemical cues trigger behavioural changes in social insects. We thus investigated whether uninfected workers in parasitized colonies are attracted by the cuticular hydrocarbons (CHCs) of infected nest-mates, which provides a first possible explanation for increase care. For this, we collected 12 parasitized colonies 83 ± 40 workers (mean ± 1 s.d.) and 12 unparasitized colonies 110 ± 25 workers (mean ± 1 s.d.) in autumn 2019. Parasitized colonies contained at least five infected ant workers. From each colony, we froze five infected and five uninfected, brood-caring workers at −20°C. CHCs were extracted by adding 0.5 ml hexane to each vial with either five nurses or five infected workers. The

**Table 1.** Results of log-rank Mantel–Cox tests comparing survival of ant categories.

| comparisons | $\chi^2$ | P | hazard ratio (95%CL) |
|---|---|---|---|
| queens from parasitized colonies–queens from unparasitized colonies | 0.34 | 0.56 | 1.27 (0.57–2.82) |
| infected workers–queens from parasitized colonies | 0.01 | 0.91 | 1.04 (0.56–1.91) |
| infected workers–queens from unparasitized colonies | 0.54 | 0.46 | 1.26 (0.68–2.34) |
| infected workers–uninfected workers from parasitized colonies | 209.1 | <0.0001 | 0.17 (0.13–0.20) |
| infected workers–uninfected workers from unparasitized colonies | 161.7 | <0.0001 | 0.18 (0.15–0.23) |
| uninfected workers from parasitized colonies–uninfected workers from unparasitized colonies | 12.65 | 0.0004 | 1.37 (1.51–1.62) |
| nurses from parasitized colonies–nurses from unparasitized colonies | 3.7 | 0.05 | 1.25 (0.98–1.57) |
| nurses from parasitized colonies–foragers from unparasitized colonies | 5.82 | 0.02 | 0.76 (0.60–0.96) |
| nurses from unparasitized colonies–foragers from unparasitized colonies | 0.42 | 0.52 | 0.93 (0.73–1.17) |
| foragers from parasitized colonies–foragers from unparasitized colonies | 10.08 | 0.002 | 1.44 (1.13–1.82) |

ants were removed after 10 min. The extracts were transferred into inlays, concentrated to approximately 15 µl by evaporating the hexane under a gentle nitrogen stream and frozen until use. A day before the experiment, we relocated 10 uninfected workers, 10 larvae and the queen of each colony in a Petri dish nest (Falcon, ⌀ 5 cm) with a plastered floor. Ten microlitres of CHC extract was added to a glass bead (Roth, ⌀ 1.7–2.1 mm), and the hexane was given 20 min to evaporate. We presented to each parasitized sub-colony two glass beads, one covered with the CHCs of uninfected nest-mates and the other one with the CHCs of infected nest-mate workers. The position of the glass beads within the nest (left or right) was randomized and they were equidistant to the centre. Each sub-colony was video-recorded for 20 min (Canon, HD Camcorder Legria HFR706). The experiments were conducted at 21°C in a climate chamber with 80% humidity. We used unparasitized colonies to investigate whether nest-mate CHCs generally elicit the interest of workers. The procedures were the same except that the sub-colonies were presented with two glass beads covered with either 10 µl of nest-mate CHC extract or 10 µl hexane. The videos were analysed with the VLC media player with the observer being blind as to the experiment and treatment of the glass beads. We noted the number of interactions directed towards each glass bead. The data were analysed with R (v. 3.5.1) using GLMM (package glm2, command glm) with a binomial or quasi-binomial distribution. The model was used to calculate whether the number of contacts towards the two alternative glass beads differed from a 50 : 50 chance. The results of one trial of a parasitized sub-colony were removed, in which the ants had little contact with the glass beads (less than 10 times). For data visualization, we used the proportion of contacts towards glass beads covered with either infected nest-mate CHCs for parasitized colonies or nest-mate CHC for parasitized colonies (command ggplot).

# 3. Results

## 3.1. Survival of uninfected and infected workers and queens

Infected workers survived considerably longer than uninfected workers (i.e. nurses and foragers), but their survival did not differ from that of queens in both parasitized and unparasitized colonies (table 1 and figure 1a). The presence of infected workers reduced the survival of their uninfected nest-mates. In particular, foragers from parasitized colonies survived less long than foragers from unparasitized colonies, while this lifespan reduction was less pronounced and borderline non-significant in nurses (table 1 and figure 1a,b). Queens from parasitized colonies showed no change in survival compared to queens from unparasitized colonies (figure 1a). Nurses in parasitized colonies survived longer than foragers, whereas nurses from unparasitized colonies survived as long as foragers (table 1 and figure 1b).

None of the 285 marked, uninfected workers in parasitized colonies survived until the end of the observation period, whereas 52 of 98 (approx. 53%) infected workers and 15 of 29 (approx. 55%) queens were still alive after 3 years. In unparasitized colonies, none of the 140 nurses, only 3 of 140

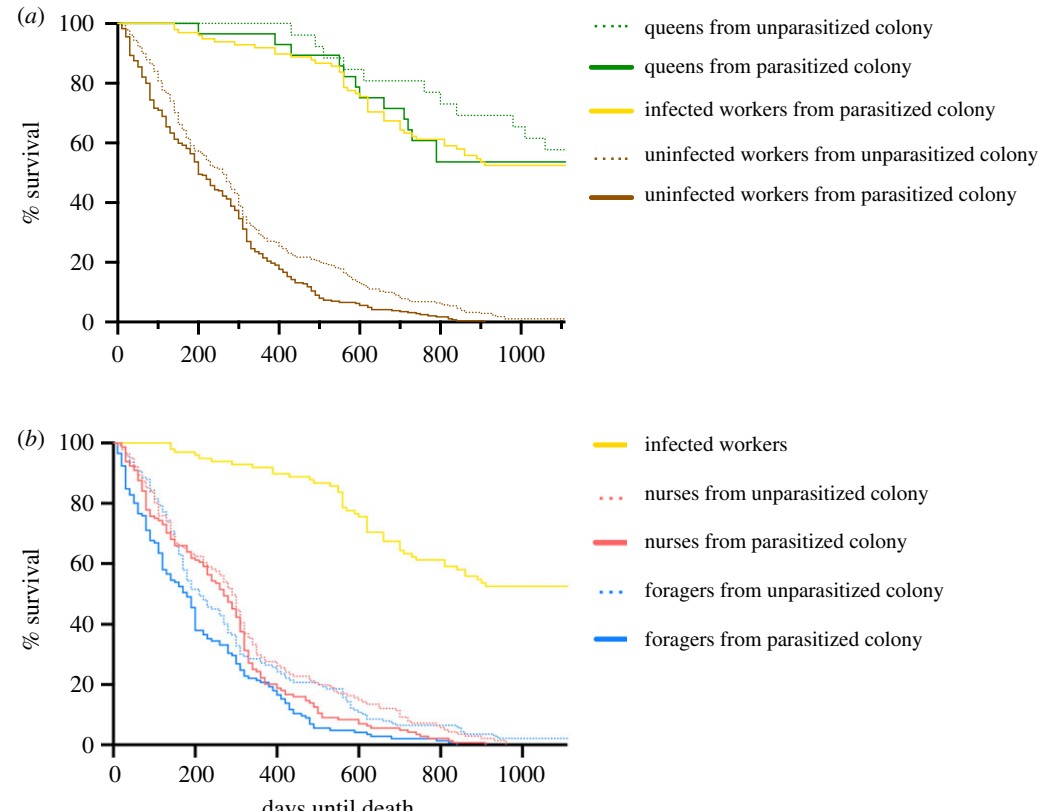

**Figure 1.** (a) Survival of queens (green), infected (yellow) and uninfected workers (brown) of T. nylanderi from unparasitized (dash) and parasitized colonies (solid) over 1110 days. (b) Survival compared between infected workers (yellow), nurses (red) and foragers (blue) from unparasitized (dash) and parasitized (solid) ant colonies. Statistical results are provided in table 1.

foragers (approx. 2%) and 17 of 28 (approx. 61%) queens were alive after 3 years. Across parasitized and unparasitized colonies, uninfected nurses survived on average for $296 \pm 216$ days, foragers for $254 \pm 217$ days, infected workers for $842 \pm 300$ days and queens for $862 \pm 309$ days (mean $\pm$ 1 s.d.). Overall, 12 of 57 (approx. 21%) colonies died within the 3-year observation period, that is both the queen and all workers perished. Parasitized colonies were as likely to die as unparasitized colonies ($n_{para} = 8$, $n_{unpara} = 4$, Fisher's exact test: $p = 0.33$).

## 3.2. Metabolic rate, body mass and lipid content of workers and queens

Infected workers and nurses exhibited similar metabolic rates ($t_{(23.35)} = 0.88$, $p = 0.39$). Compared with all other categories, queens had a lower (all $p < 0.002$) and foragers a higher metabolic rate (all $p < 0.006$; figure 2a). Infected workers and nurses also had a similar body mass ($t_{(25.36)} = 1.36$, $p = 0.19$) and relative lipid content, although there was a trend for infected workers to contain a lower fraction of lipids ($t_{(25.32)} = 1.51$, $p = 0.07$; figure 2b,c). Queens had the highest body mass, while foragers had the lowest, yet both had similar relative lipid contents ($t_{(18.99)} = 0.39$, $p = 0.70$). Infected workers did not differ from queens and foragers in their relative fat contents ($t_{(22.99)\ \text{queens}} = 1.09$, $p = 0.29$, $t_{(21.97)\ \text{foragers}} = 1.46$, $p = 0.16$; figure 2c). Ants from parasitized and unparasitized colonies did not differ in their metabolic rate, body mass and lipid content (table 2).

## 3.3. Social care provided to workers and queens

In parasitized colonies, social care behaviours directed towards infected workers exceeded those towards nurses, foragers and even the queen (table 3 and figure 2d). Active begging behaviour for food was very rarely observed (5 out of 1780 total interactions), and infected workers were never involved in these interactions. Foragers received less care than nurses and nurses received less care than the queen

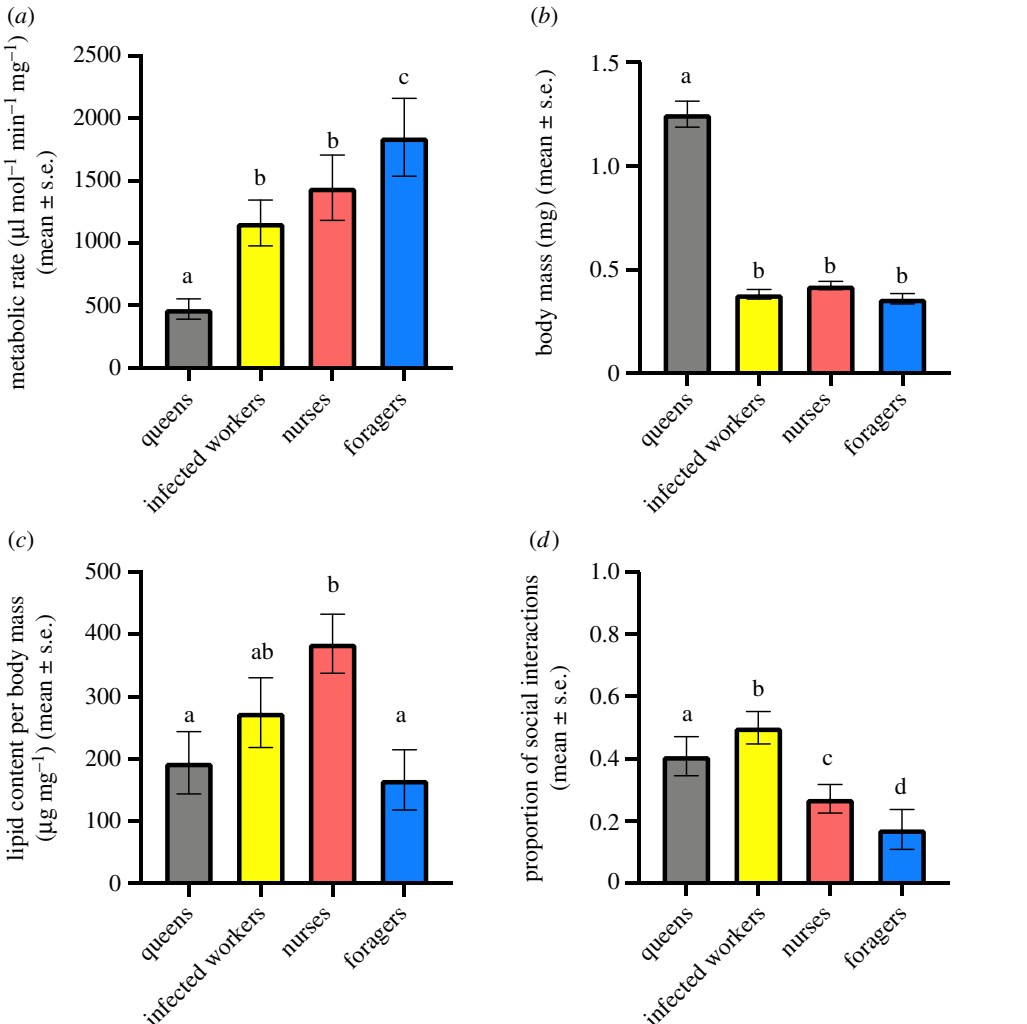

**Figure 2.** Differences in (*a*) metabolic rate, (*b*) body mass, (*c*) lipid content and (*d*) received social care of queens (grey), infected workers (yellow), nurses (red) and foragers (blue) from *T. nylanderi* colonies. Vertical lines represent standard error. Different letters indicate significant differences between categories. Statistical results are provided in tables 2 and 3.

**Table 2.** Results of LMM comparing metabolic rate, lipid content and body mass measurements ant categories and colony types.

| comparisons | metabolic rate | | lipid content | | body mass | |
|---|---|---|---|---|---|---|
| | $\chi^2$ | $p$ | $\chi^2$ | $p$ | $\chi^2$ | $p$ |
| infected workers–nurses–foragers–queens of parasitized colonies | 96.62 | <0.0001 | 41.61 | <0.0001 | 551.1 | <0.001 |
| parasitized–unparasitized colonies | 0.02 | 0.89 | 0.14 | 0.15 | 0.02 | 0.89 |
| interaction: ant category–colony parasitism status | 0.58 | 0.75 | 1.25 | 0.54 | 0.58 | 0.75 |

(table 1 and figure 2*d*). The rate of social care was unaffected by colony parasitism status as a main effect nor in interaction with individual ant category (table 3).

## 3.4. Response towards cuticular hydrocarbons of infected and uninfected workers

In unparasitized colonies, significantly more attention was directed towards nest-mate-derived CHCs, relative to the hexane control (GLM: $z = 3.07$, $p = 0.002$, figure 3*b*). In parasitized colonies, the CHCs from infected nest-mates elicited more responses than the CHCs from uninfected nest-mates (GLM:

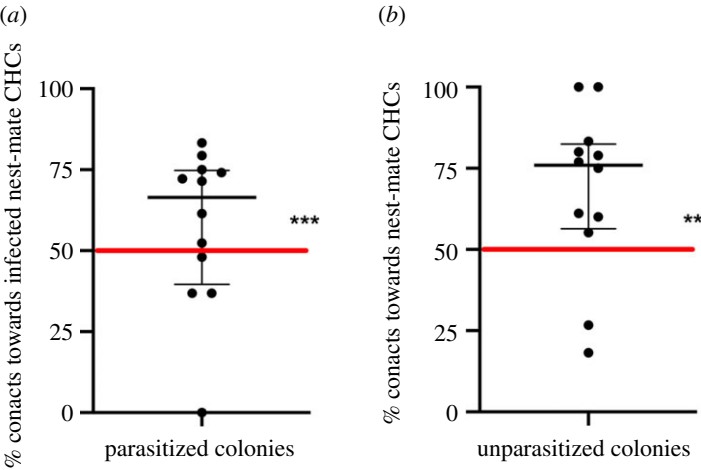

**Figure 3.** Response of ants to glass beads covered with cuticular hydrocarbon extracts. (*a*) Contacts of uninfected workers from parasitized colonies to glass beads covered with CHCs of infected nest-mates to an uninfected nest-mate control. (*b*) Contacts of uninfected workers from unparasitized colonies to glass beads covered with CHCs of nest-mate workers to a hexane control.

**Table 3.** Results of GLMM comparing the frequency of social behaviour between different ant categories and colony types.

| Comparison | $\chi^2$/z | p |
|---|---|---|
| parasitized colonies | | |
| infected workers–nurse–forager–queens | $\chi^2 = 75.49$ | <0.0001 |
| infected workers–nurses | $z = 5.76$ | <0.0001 |
| infected workers–foragers | $z = 8.03$ | <0.0001 |
| infected workers–queens | $z = 2.65$ | 0.008 |
| foragers–nurses | $z = -3.16$ | 0.002 |
| nurses–queens | $z = -2.99$ | 0.002 |
| parasitism status of colony | $\chi^2 = 0.03$ | 0.87 |
| interaction ant category–parasitism status | $\chi^2 = 2.85$ | 0.24 |

$z = 6.67$, $p < 0.0001$; figure 3*a*), suggesting that the chemical profile of infected workers is more attractive for nest-mates.

## 4. Discussion

Parasites generally reduce the fitness of their hosts. In those rare cases in which parasites extend their hosts' lifespan, they typically decrease the fecundity of their hosts [20,21]. Here, we report an at least threefold prolonged lifespan of ant workers infected with a helminth, and these infected workers exhibit a similar if not larger reproductive potential than uninfected members of their caste [24]. During our 3-year observation period, the survival of infected workers was similar to that of queens, which can live for up to two decades in this species [39]. The observed differences in survival were extreme. While more than half of all infected workers were still alive after more than 1000 days, all of their uninfected nest-mate workers had already died.

Endoparasites are scavengers of host resources. Infection with tapeworm larvae may therefore cause a reduction in body mass and fat content, and possibly an increase in metabolic rate of host workers. Alternatively, as infected workers survive as long as the queen does, they might resemble the royal caste also in physiology. We found more support for the first hypothesis as infected workers exhibited a similar metabolic, body mass and lipid content as nurses do, which are the youngest members of ant colonies [36]. Infected workers also resemble nurses in their cuticular hydrocarbon profile and are able to invest in the development of their ovaries in the absence of the queen, just like many

nurses do [24,36]. In addition, infected workers are often inactive and stay in close contact with the brood, which is the typical location of nurses within an ant nest [24,26,37].

In many traits, infected workers thus appear to age more slowly, but what are the proximate causes for their extended lifespan? Presently, we are lacking mechanistic explanations and multiple factors including intrinsic physiological changes and extrinsic conditions seem to play a role [40–43]. Whether and how social behaviours contribute to lifespan differences in social insects is unknown. We show that infected workers not only obtain more care than the average uninfected workers, but also more than the queen, which is usually the best cared for ant in the colony. Could such differences indirectly modulate lifespan and if yes, what are the underlying mechanisms? Studies addressing these questions will be the scope of future studies.

Infected workers are well provided with food [26], and our behavioural observations in this study do not indicate that they actively beg for food more than other workers. Rather, our experiments with cuticular hydrocarbon extracts indicate that the chemical signals of infected workers are highly attractive for their nest-mates. It is not surprising that infected workers signal their needs in this way as communication in ant societies is primarily mediated through chemical cues [44]. The increased attention directed towards infected workers might be caused by specific cuticular compounds, or simply by their profile being different from that of other workers [28]. The latter explanation is less likely as we could previously show that the chemical profile of infected workers is closer to that of uninfected nurses, than the latters' CHC profile is to that of foragers [31]. It remains unclear, whether and how the infection changes chemical signalling in infected workers.

Being infected by a parasite is usually costly for the host. In this host–parasite system, there are no direct negative effects on the infected workers themselves; rather the cost of parasitism becomes visible in their uninfected nest-mates. Indeed, uninfected foragers of parasitized colonies exhibit elevated mortality compared with workers residing in unparasitized colonies, as shown here and in an earlier study [23]. This increased mortality was apparent, although our ant colonies were well maintained in our laboratory, regularly receiving ample food and water. Potentially, the care and high food demands of infected workers increase the workload of their nest-mates, which in turn might cause their increased mortality [26]. Yet, our physiological data did not point to increased physiological stress, as infected workers in parasitized colonies exhibited a similar metabolic rate, body mass and lipid content to those from unparasitized colonies. Moreover, the survival of queens from parasitized colonies was unaffected by colony parasitism. The well-being of the queen is critical for the entire colony as only she can ensure long-term colony survival. Thus, though workers from parasitized colonies provide more care for their infected nest-mates, they should avoid neglecting the queen. Preliminary observations indicate a decrease in queen care with an increasing fraction of infected workers (A. Lenhart 2020, personal observation). Moreover, parasitized colonies raise more queen–worker intercastes and show a male-biased sex ratio, both are potential signals of stress on the colony level [26]. However, these negative consequences of parasitism are rather weak as *A. brevis* does not negatively affect the reproductive output of *T. nylanderi* field colonies and parasitized colonies are as often queenright as unparasitized colonies [26].

Parasites with complex life cycles regularly alter the phenotype of their intermediate hosts [45,46]. Besides changes in life history, the behaviour and morphology of hosts are often profoundly modified [47]. Such alterations might represent host compensatory responses or side effects of infection, but could also be adaptations of the parasite to secure survival or enhance transmission [46]. Workers of *T. nylanderi* ants infected by the tapeworm *A. brevis* are highly modified organisms. Next to their extraordinary long lifespan, the infection reduces the behavioural repertoire of the workers [24,26]. They remain predominantly inactive inside the nest and show reduced anti-predatory responses [23,26]. This stands in stark contrast to behavioural changes observed in ants used as intermediate hosts by other parasites with complex life cycles. Ant workers infected by the lancet liver fluke leave the nest to expose themselves to the definitive host of this parasite, large herbivore mammals [48]. *Temnothorax* workers are tiny, about 2–3 mm in length, and live in the leaf litter, where they are difficult to spot for their definitive hosts, woodpeckers [25]. Sending infected workers out of the nest and exposing them to high extrinsic mortality might be a rather unsuccessful strategy. Instead, infected workers remain in the safety of their acorn or stick nests, which is the very place woodpeckers are opening in search of insect larvae. Thus, it is likely that these behavioural alterations observed in tapeworm-infected workers actually may predispose ants to predation by birds. Accordingly, the extended lifespan could be caused by the parasite in order to prolong the period of possible transmission. The lifespan extension might be especially feasible in ant workers, which can rely on the care from their nest-mates and where genes for a long life, that is queen-specific genes,

could be activated [30]. Lifespan extension following infection with parasites has been found in other host–parasite systems [20,22,49–52]. In these cases, the parasite either reduces the fecundity of its host thereby triggering a lifespan extension or it appears to increase host survival to be able to finish its own development before the host dies. Our case lays somewhat different, as here the parasite may benefit from a lifespan extension of its intermediate host to facilitate its transmission to the final host.

Ethics. Ant collection permits were obtained from local forestry departments. We followed the guidelines of the Study of Animal Behaviour and the legal and institutional rules.

Data accessibility. All raw data presented here has been uploaded in Dryad: https://doi.org/10.5061/dryad.1jwstqjt6 [53].

Authors' contributions. S.B., F.M. and S.F. designed the experiments. S.B., A.L. and M.A.N. performed the experiments. S.B. carried out the statistical analyses. All authors contributed to writing the manuscript.

Competing interests. We declare no competing interests.

Funding. We thank the German Research Foundation for funding this research project (DFG; grant nos. FO 298/15-1 and ME 2842/3-1).

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
