## [Peer Review File · Royal Society Open Science]

Review History

RSOS-202118.R0 (Original submission)

Review form: Reviewer 1

Is the manuscript scientifically sound in its present form?

Yes

Are the interpretations and conclusions justified by the results?

Yes

Is the language acceptable?

Yes

Do you have any ethical concerns with this paper?

No

Have you any concerns about statistical analyses in this paper?

No

Recommendation?

Accept with minor revision (please list in comments)

Comments to the Author(s)

The authors have continued to make changes to the manuscript since its prior submission to Proceedings B, that have improved it and add interesting insights. In particular, the addition of data on interactions of nest mates with extracts CHCs of infected and uninfected ants provides at least a primate mechanism for understanding increased social interactions to infected ants. My remaining comments are minor and are mainly for clarity.

Line 21-23: The two starting sentences don't make sense together. The first is about the evolution of host phenotypes and the second is the impact of parasitism on host phenotypes. Given it is the focus of the manuscript, I recommend on keeping just the latter.

Line 32: Rather, "elicit more interest from..."

Line 56: It is unclear how the paragraph above has "highlighted the importance of parasites for the evolution of host phenotype." I would recommend, "This highlights the importance of parasites for social insect host phenotypes and reveals how social traits can be intertwined with parasite-induced alterations."

Line 58: Rather, "Parasite infected hosts generally incur fitness costs, such as..."

Line 69: Remove comma - "...is completed when woodpeckers"

Line 77-78: Following up from earlier comments by the second referee, the authors don't actually investigate if superior care is provided and if this directly affects longevity. This would need manipulation of provided care in both groups. Therefore, I recommend rewording this sentence.

Line 84-85: I don't think "due to a younger age" is what is intended here. Perhaps rather, "traits more typically expressed by younger ants"?

Line 138: Rather, "We collected an additional..."

Line 204: Brood care? Perhaps brood caring?

Results: Often statements are made comparing infected workers and nurses. I believe the latter are also uninfected. If this is correct this should be explicitly stated and also the wording modified as nurses are also workers.

Line 282-285: I would recommend switching these sentences and elaborating what the connection is. For example, "In unparasitized colonies we show that significantly more attention is directed towards nestmate-derived CHCs, relative to a control (statistics). Uninfected workers of parasitized colonies show greater interactions with parasitized nestmate CHCs relative to CHCs from non-parasitized nestmates (statistics)."

Line 288: Rather, "In those rare cases in which they extend their host's lifespan, they..."

Review form: Reviewer 2

Is the manuscript scientifically sound in its present form?

Yes

Are the interpretations and conclusions justified by the results?

Yes

Is the language acceptable?

Yes

Do you have any ethical concerns with this paper?

No

Have you any concerns about statistical analyses in this paper?

No

Recommendation?

Accept with minor revision (please list in comments)

Comments to the Author(s)

This is an interesting paper about an interesting phenomenon. The authors make their case effectively and the paper reads well. I have no major suggestions to make, but here are some minor ones.

Line 211 needs rephrasing. I suggest, " CHC extract-covered glass beads on metal plates".

Paragraph beginning at line 230. These statistics would be better presented as a table. It is hard to keep track in the present format.

Line 256 onward. Please provide the means along with the statistics.

Line 306. Other examples of workers aging at different rates come to mind. Overwintering honeybees age more slowly than summer bees. The same is true for the overwintering workers of harvester ants compared to those born in the summer (see papers by Kwapich). Aging seems to be under some kind of physiological control. External behavioral control seems less likely to me.

Line 509 and 510: "contacts", not "contracts".

My most important suggestion is to label the lines in Fig. 1 directly, rather than referencing them in the caption. Or do both, but following the present version is tedious.

The y-axis in Fig. 3 is proportion, not %.

Decision letter (RSOS-202118.R0)

Dear Dr Foitzik

On behalf of the Editors, we are pleased to inform you that your Manuscript RSOS-202118 "Extreme lifespan extension in tapeworm-infected ant workers" has been accepted for publication in Royal Society Open Science subject to minor revision in accordance with the referees' reports. Please find the referees' comments along with any feedback from the Editors below my signature.

Please submit your revised manuscript and required files (see below) no later than 7 days from today's (ie 26-Mar-2021) date. Note: the ScholarOne system will 'lock' if submission of the revision is attempted 7 or more days after the deadline. If you do not think you will be able to meet this deadline please contact the editorial office immediately.

on behalf of Professor Leslie Brown (Associate Editor) and Kevin Padian (Subject Editor)
openscience@royalsociety.org

Associate Editor Comments to Author (Professor Leslie Brown):

The paper was reviewed by two reviewers and only minor changes are recommended. Please affect these changes since it will improve the quality and value of your paper. We appreciate your effort in writing and submitting the paper. Please indicate that the changes have been affected and if not please provide a motivation for it.

Reviewer comments to Author:

Reviewer: 1

Comments to the Author(s)

The authors have continued to make changes to the manuscript since its prior submission to Proceedings B, that have improved it and add interesting insights. In particular, the addition of data on interactions of nest mates with extracts CHCs of infected and uninfected ants provides at least a primate mechanism for understanding increased social interactions to infected ants. My remaining comments are minor and are mainly for clarity.

Line 21-23: The two starting sentences don't make sense together. The first is about the evolution of host phenotypes and the second is the impact of parasitism on host phenotypes. Given it is the focus of the manuscript, I recommend on keeping just the latter.

Line 32: Rather, "elicit more interest from..."

Line 56: It is unclear how the paragraph above has "highlighted the importance of parasites for the evolution of host phenotype." I would recommend, "This highlights the importance of parasites for social insect host phenotypes and reveals how social traits can be intertwined with parasite-induced alterations."

Line 58: Rather, "Parasite infected hosts generally incur fitness costs, such as..."

Line 69: Remove comma - "...is completed when woodpeckers"

Line 77-78: Following up from earlier comments by the second referee, the authors don't actually investigate if superior care is provided and if this directly affects longevity. This would need manipulation of provided care in both groups. Therefore, I recommend rewording this sentence.

Line 84-85: I don't think "due to a younger age" is what is intended here. Perhaps rather, "traits more typically expressed by younger ants"?

Line 138: Rather, "We collected an additional..."

Line 204: Brood care? Perhaps brood caring?

Results: Often statements are made comparing infected workers and nurses. I believe the latter are also uninfected. If this is correct this should be explicitly stated and also the wording modified as nurses are also workers.

Line 282-285: I would recommend switching these sentences and elaborating what the connection is. For example, "In unparasitized colonies we show that significantly more attention is directed towards nestmate-derived CHCs, relative to a control (statistics). Uninfected workers of parasitized colonies show greater interactions with parasitized nestmate CHCs relative to CHCs from non-parasitized nestmates (statistics)."

Line 288: Rather, "In those rare cases in which they extend their host's lifespan, they..."

Reviewer: 2

Comments to the Author(s)

This is an interesting paper about an interesting phenomenon. The authors make their case effectively and the paper reads well. I have no major suggestions to make, but here are some minor ones.

Line 211 needs rephrasing. I suggest, "CHC extract-covered glass beads on metal plates".

Paragraph beginning at line 230. These statistics would be better presented as a table. It is hard to keep track in the present format.

Line 256 onward. Please provide the means along with the statistics.

Line 306. Other examples of workers aging at different rates come to mind. Overwintering honeybees age more slowly than summer bees. The same is true for the overwintering workers of harvester ants compared to those born in the summer (see papers by Kwapich). Aging seems to be under some kind of physiological control. External behavioral control seems less likely to me.

Line 509 and 510: "contacts", not "contracts".

My most important suggestion is to label the lines in Fig. 1 directly, rather than referencing them in the caption. Or do both, but following the present version is tedious.

The y-axis in Fig. 3 is proportion, not %.

===PREPARING YOUR MANUSCRIPT===

===PREPARING YOUR REVISION IN SCHOLARONE===

Author's Response to Decision Letter for (RSOS-202118.R0)

See Appendix A.

Decision letter (RSOS-202118.R1)

Dear Dr Foitzik,

It is a pleasure to accept your manuscript entitled "Extreme lifespan extension in tapeworm-infected ant workers" in its current form for publication in Royal Society Open Science. The comments of the reviewer(s) who reviewed your manuscript are included at the foot of this letter.

on behalf of Professor Leslie Brown (Associate Editor) and Kevin Padian (Subject Editor)
openscience@royalsociety.org

Associate Editor Comments to Author (Professor Leslie Brown):

Thank you very much for the resubmission and the corrections made and the detailed responses you provided to the reviewer's' comments and suggestions.

JOHANNES GUTENBERG-UNIVERSITÄT MAINZ - 55099 Mainz

02 April 2021

Resubmission of revised manuscript ID RSOS-202118

Dear Lianne Parkhouse,

We would like to resubmit our revised ms „*Extreme lifespan extension in tapeworm-infected ant workers*“ to *Royal Society Open Science*. We were very happy by the overall positive response to our ms and the acceptance with minor revision. We have altered the ms as suggested by the reviewers and outline below in detail how we followed their recommendations. We hope with these final changes, the paper can now be accepted. All co-authors agreed to this submission. If you require any additional information, please do not hesitate to contact me.

Yours Sincerely,

Prof. Dr. Susanne Foitzik

FACULTY 10
BIOLOGY

iomE

Institute of Organismic
and Molecular Evolution

Univ. Prof.
Dr. Susanne Foitzik

Behavioral Ecology and
social Evolution

Johannes Gutenberg-
University Mainz

Hanns-Dieter-Hüsch-Weg 15
D-55128 Mainz

Tel. +49 6131 39-27840
Fax +49 6131 39-27850

foitzik@uni-mainz.de
www.uni-mainz.de

2

Reviewer comments to Author:

Reviewer: 1

Comments to the Author(s)

The authors have continued to make changes to the manuscript since its prior submission to Proceedings B that have improved it and add interesting insights. In particular, the addition of data on interactions of nest mates with extracts CHCs of infected and uninfected ants provides at least a primate mechanism for understanding increased social interactions to infected ants. My remaining comments are minor and are mainly for clarity.

Thanks for the overall positive review. We conducted the changes as recommended.

Line 21-23: The two starting sentences don't make sense together. The first is about the evolution of host phenotypes and the second is the impact of parasitism on host phenotypes. Given it is the focus of the manuscript, I recommend on keeping just the latter.

The new line 21 reads: "Social insects are hosts of diverse parasites, but the influence of these parasites on phenotypic host traits is not yet well understood."

Line 32: Rather, "elicit more interest from..."

The new line 30 reads: "This increased attention could be mediated by their deviant chemical profile, which we determined to elicit more interest from uninfected nestmates in a separate experiment."

Line 56: It is unclear how the paragraph above has "highlighted the importance of parasites for the evolution of host phenotype." I would recommend, "This highlights the importance of parasites for social insect host phenotypes and reveals how social traits can be intertwined with parasite-induced alterations."

The new line 54 reads: "This highlights the importance of parasites for the phenotypes of social insect hosts and reveals how social traits can be intertwined with parasite-induced alterations."

Line 58: Rather, "Parasite infected hosts generally incur fitness costs, such as..."

3

The new line 57 reads: "Parasite infected hosts generally incur fitness costs, such as a slower development, reduced survival, or a lower fecundity (19)."

Line 69: Remove comma - "...is completed when woodpeckers"

Corrected.

Line 77-78: Following up from earlier comments by the second referee, the authors don't actually investigate if superior care is provided and if this directly affects longevity. This would need manipulation of provided care in both groups. Therefore, I recommend rewording this sentence.

The reviewers are pointing out a legitimate limitation of our study, as we have indeed not performed any experimental manipulations to investigate superior care nor further examined whether social care can prolong lifespan. We therefore carefully revised our phrasing throughout the manuscript.

For example, the following changes were made for the new lines 81-93: "The factors that directly or indirectly regulate lifespan in social insects are still poorly understood. In particular, the role of social behaviors of group members on longevity has hardly been studied so far. In our focal species, infected workers are known to be fed more often than their nestmates (26). We thus were interested to gain insights in whether ants in parasitized societies receive more social attention depending on their infection status or caste. We further examined whether the deviant cuticular hydrocarbon profile of infected workers is more attractive to their nestmates, which could explain why they receive more attention (28,30). Physiological traits are known to be associated with the pace of life and could therefore be good markers of longevity (31-33). We thus finally analyzed the metabolic rate and lipid content of workers and queens in parasitized and unparasitized colonies."

Line 84-85: I don't think "due to a younger age" is what is intended here. Perhaps rather, "traits more typically expressed by younger ants"?

The new line 82 reads: "We were interested in how long infected workers can live or whether their reported increase in survival might be due to traits more typically expressed by younger ants."

Line 138: Rather, "We collected an additional..."

Corrected.

Line 204: Brood care? Perhaps brood caring?

4

The new line 203 reads: "From each colony, we froze five infected and five uninfected, brood caring workers at -20°C."

Results: Often statements are made comparing infected workers and nurses. I believe the latter are also uninfected. If this is correct this should be explicitly stated and also the wording modified as nurses are also workers.

Indeed, all nurses and foragers were chosen based on their behaviour/position and their body coloration – which is a reliable sign for being not infected with *A. brevis*. We mention early in the methods that when we refer to nurses & foragers, we speak of (young) brood-caring, uninfected workers & (old) foraging, uninfected workers.

The following sentences were rephrased and added to the current draft.

New line 102: "Young workers - so-called callows - are easily distinguishable from older ones by their light, not yet fully sclerotized cuticle. These workers focus on brood care and are henceforth referred to as nurses. Workers leaving the nest to forage for food – so-called foragers - are usually older and were at the time of collection likely one year or older (34,35)."

New line 109: "All nurses and foragers were identified to be uninfected based on their body colouration (e.g. brownish, with abdominal stripe) in this and all following experiments."

Line 282-285: I would recommend switching these sentences and elaborating what the connection is. For example, "In unparasitized colonies we show that significantly more attention is directed towards nestmate-derived CHCs, relative to a control (statistics). Uninfected workers of parasitized colonies show greater interactions with parasitized nestmate CHCs relative to CHCs from non-parasitized nestmates (statistics)."

We rephrased the sentence in the new line 282 as follows: "In unparasitized colonies, significantly more attention was directed towards nestmate-derived CHCs, relative to the hexane control (GLM: $z = 3.07$, $p = 0.002$, Fig. 3b). In parasitized colonies, the CHCs from infected nestmates elicited more responses than the CHCs from uninfected nestmates (GLM: $z = 6.67$, $p < 0.0001$, Fig. 3a), suggesting that the chemical profile of infected workers is more attractive for uninfected nestmates. "

Line 288: Rather, "In those rare cases in which they extend their host's lifespan, they..."
The new line 289 reads: "In those rare cases in which parasites extend their hosts' lifespan, they typically reduce their hosts' fecundity (20,21)."

5

Reviewer: 2

Comments to the Author(s)

This is an interesting paper about an interesting phenomenon. The authors make their case effectively and the paper reads well. I have no major suggestions to make, but here are some minor ones.

Line 211 needs rephrasing. I suggest, "CHC extract-covered glass beads on metal plates".

The new line 211 reads: "We presented to each parasitized sub-colony two glass beads, one covered with the CHCs of uninfected nestmates and the other one with the CHCs of infected nestmate workers."

Paragraph beginning at line 230. These statistics would be better presented as a table. It is hard to keep track in the present format.

We have now included three tables reporting the test statistics to not disturb the flow of reading.

Line 306. Other examples of workers aging at different rates come to mind. Overwintering honeybees age more slowly than summer bees. The same is true for the overwintering workers of harvester ants compared to those born in the summer (see papers by Kwapich). Aging seems to be under some kind of physiological control. External behavioral control seems less likely to me.

Thank you for bringing up other interesting examples of lifespan plasticity in social insects – we included the work by Kwapich and two other references to our manuscript (new references 40-42). Indeed, there are a few indications pointing to physiological factors modulating lifespan in social insects, yet the mechanistic explanations are lacking. Social care could include providing workers with more food as parasites scavenge essential nutrients. It is known that the diet composition can affect lifespan in workers (work by e.g. Dussutour). It has been also shown that during oral transfers, workers exchange all kind of metabolites, which could alter physiological process – potentially also those ones regulating lifespan? These are interesting questions for future studies. Given that social care differs in parasitized colonies depending on the infection status of workers, we briefly point out that social behaviours could potentially be indirect factors influencing lifespan.

The new line 300 in the discussion reads: "In many traits, infected workers thus appear to age more slowly, but what are the proximate causes for their extended lifespan? Presently, we are lacking mechanistic explanations and multiple factors including intrinsic physiological changes and extrinsic conditions seem to play a role (40-42). Whether and how social behaviours contribute to

6

lifespan differences in social insects is unknown. We show that infected workers not only obtain more care than the average uninfected workers, but also more than the queen, which is usually the best cared for ant in the colony. Could such differences indirectly modulate lifespan and if yes, what are the underlying mechanisms? Studies addressing these questions will be the scope of future studies.“

Line 509 and 510: "contacts", not "contracts".

Corrected.

My most important suggestion is to label the lines in Fig. 1 directly, rather than referencing them in the caption. Or do both, but following the present version is tedious.

We kept the figure legend, but added a caption into Fig 1a. & Fig 1b.

7

The y-axis in Fig. 3 is proportion, not %.

Corrected. See Figures attached.

(a)

(b)